# Urethral Mesh Assessment in Cancer Patients

**DOI:** 10.3390/cancers15235599

**Published:** 2023-11-27

**Authors:** Roxana Pintican, Anne Buckley, Diana Feier, Susan Freeman

**Affiliations:** 1Department of Radiology, “Iuliu Hatieganu” University of Medicine and Pharmacy, 400347 Cluj-Napoca, Romania; roxana.pintican@gmail.com; 2Department of Radiology, Cambridge University Hospitals NHS Foundation Trust, Cambridge CB2 0QQ, UK; bucklear@tcd.ie (A.B.); sue.freeman@addenbrookes.nhs.uk (S.F.)

**Keywords:** mesh, urethral mesh, TOT, TVT, urethral neoplasm

## Abstract

**Simple Summary:**

This suggested paper aims to comprehensively address the assessment of urethral mesh in oncologic patients, a common surgical intervention for stress urinary incontinence. Acknowledging the procedure’s benefits along with its potential complications, we seek to present a comprehensive review encompassing normal MR and CT appearances of urethral mesh, as well as pathological aspects tied to complications, enhancing patient outcomes and fostering an informed decision-making within the field of urethral mesh management for oncologic patients.

**Abstract:**

Urethral mesh placement has become a common surgical intervention for the management of stress urinary incontinence. While this procedure offers significant benefits, it is not without potential complications. This review article aims to provide a comprehensive overview of urethral mesh assessment in oncologic patients. The article explores normal magnetic resonance imaging (MRI) and computed tomography (CT) mesh appearances and highlights the pathological aspects associated with urethral mesh complications including both short-term and long-term post-operative complications. By understanding the spectrum of normal findings of urethral mesh and the possible complications, clinicians can improve patient outcomes and make informed decisions regarding urethral mesh management in this patient population.

## 1. Introduction

The history of urethral meshes traces a journey from ancient civilizations’ use of natural materials like animal ligaments to reinforce weakened tissues, to the mid-20th-century breakthrough of synthetic meshes, notably polypropylene, revolutionizing urological surgery, especially in treating stress urinary incontinence (SUI) [1]. These synthetic mesh tapes, while widely used, posed challenges due to proinflammatory responses after implantation [2]. The advent of the first generation tension-free vaginal tape (TVT) marked a significant milestone, yet blind access during TVT procedures led to complications, necessitating routine cystoscopy [3]. In 2001, second-generation slings emerged, like transobturator meshes (TOT), offering safer alternatives [4]. The latest innovation, the mini sling or single-incision sling (SIS), introduced a minimally invasive approach, involving a single vaginal incision, reducing surgical invasiveness, potentially improving recovery, and minimizing post-operative discomfort [5,6].

Nowadays it is generally accepted that urethral mesh placement has revolutionized the management of stress urinary incontinence (SUI) in female patients, providing a minimally invasive and effective solution [7]. The last two decades have seen a marked increase in the number of SUI procedures, and therefore in the number of mesh placement procedures [8]. However, this procedure is not without potential complications, with studies reporting a cumulative incidence of subsequent sling revision or removal for various indications of 3.7% [9]. 

Although mesh surgery does not increase the risk of cancer [10], women may still develop pelvic malignancy, or require urethral mesh secondary to cancer surgery. MRI has an emerging role in evaluating the female pelvis in oncological diseases, allowing pre-operative staging, risk stratification and treatment selection of the patients [11,12]. MRI provides optimal visualization of the mesh and adjacent anatomical structures enabling accurate assessment and diagnosis of complications. The follow-up of patients with cancer is usually undertaken with MRI or CT imaging and less frequently abdominal ultrasound or conventional radiology [12,13,14].

Mesh-related complications can arise from various factors, including surgical technique, patient-specific factors, and the interaction between the mesh and the surrounding tissues. Short-term (intraoperative and/or immediate post-operative) and long-term, delayed complications (weeks or months after the initial surgery) may arise [15]. In order to optimize patient outcomes, it is crucial for clinicians to have a comprehensive understanding of the potential complications and the available treatment options.

Existing guidelines recognize the usefulness of imaging methods in identifying and characterizing urethral mesh complications [16], although they do not specify which imaging method should be used as a first line imaging tool. Ultrasound and dedicated MRI should be considered. In our institution, non-oncology patients with symptoms suggestive of urethral mesh complications will be referred for ultrasound. If this is not able to fully characterize the suspected urethral lesion, patients are referred for MRI. 

In the case of oncological, symptomatic patients, MRI is used of first intention in our institution. A dedicated MR is crucial in assessing the exact site and extent of mesh related complications. Particularly in patients with known exposed or extruded mesh, MRI provides information regarding the remainder of the mesh that is inaccessible to clinical examination or ultrasound.

In this review, we will highlight the normal appearance of urethral mesh in oncological patients, together with short-term and long-term complications associated with mesh placement. The focus will be on MRI as the main imaging modality used in pelvis-related cancers, highlighting its potential benefit in guiding treatment decisions and improving patient care. 

## 2. Types of Urethral Meshes

Over the years, several types of urethral meshes have been developed to address SUI in female patients. These meshes have evolved in terms of design, materials, and surgical techniques. 

Tension-Free Vaginal Tape (TVT) was introduced in the late 1990s, and was one of the first meshes used for the treatment of SUI [3]. It consists of a narrow synthetic mesh tape that is placed under the mid-urethra, with a retropubic route to provide support. TVT has shown good long-term success rates and has become one of the most commonly used meshes for SUI [3,17].

Transobturator meshes (TOT) were developed in the early 2000s, as an alternative to TVT [4]. TOT meshes are inserted through the obturator foramen, providing support to the mid-urethra and bladder neck. TOT meshes were designed to potentially reduce the risk of bladder or bowel injuries associated with TVT [3] (Figure 1).

As a modification of the TVT and TOT meshes, mini-slings were introduced in the mid-2000s [18]. These smaller mesh tapes require less dissection and are considered less invasive. However, even if mini-slings aim to achieve similar success rates, their short- and long-term efficacy and complications have yet to be established [4,19].

## 3. Normal Appearance of Urethral Meshes on MRI/CT

MRI plays a crucial role in evaluating the placement and integrity of urethral meshes, assessing their positioning, mesh shape, signal intensity on T1WI and T2WI sequences, and post-contrast aspects.

A properly placed TVT mesh appears as a thin, linear structure extending posterior to the mid-urethra and pubic bone towards the lower anterior abdominal wall. It typically demonstrates low signal intensity on both T1 and T2WI sequences due to its synthetic composition. In the absence of complications, the mesh should have a smooth contour, without any kinks or folds and maintain its position relative to the adjacent structures. Post-contrast MRI typically shows no significant enhancement within the mesh, indicating its inert nature (Figure 2 and Figure 3).

TOT meshes have a distinct configuration compared to TVT meshes. They consist of two arms that extend laterally from the posterior aspect of the mid-urethra, passing through the obturator foramen. On MRI, TOT meshes appear as linear structures with additional lateral arms. The mesh typically exhibits low signal intensity on both T1 and T2WI, similar to TVT meshes but in a hammock-shaped manner. Post-contrast MRI shows no significant enhancement within the mesh (Figure 4).

Some cancer patients tend to be followed up by CT, which is why it is necessary to recognize and assess the urethral mesh and, if necessary, refer for further MR evaluation. On non-contrast CT images, TVT and TOT meshes appear as thin, linear structures in the pelvis, with low attenuation due to their synthetic composition [20]. The mesh should be well-positioned under the mid-urethra and extending towards anterior abdominal wall (TVT) or laterally through the obturator foramen (TOT). The mesh should be free of any kinks, folds, or displacement. The absence of any significant surrounding soft tissue abnormalities indicates a normal appearance. CT with intravenous contrast administration can provide additional information about the vascularity of the surrounding tissues and potential complications. A normal TVT or TOT mesh should show minimal or no enhancement on post-contrast CT images. The lack of significant contrast uptake within the mesh indicates its inert nature and lack of complication. It is important to note that the presence of avid contrast enhancement within the mesh may indicate an abnormality, such as infection or inflammation (Figure 5).

## 4. Urethral Mesh in Oncological Patients

There are several types of malignancies that may involve the pelvic structures, such as gynecological cancer (ovarian, cervical, endometrial), rectal or bladder cancer and other rarer causes and mimickers. For ease of reading, we will refer to all of them below as “pelvic cancers”. Surgical techniques along with radiotherapy and chemotherapy have increased the survival of patients with pelvic cancers, while increasing the importance of their quality of life (QOL) [21,22,23,24]. 

Urinary incontinence and overactive bladder are both pathological conditions that have a high prevalence after pelvic surgery and are known to greatly impair QOL. Urinary incontinence particularly affects women, causing adverse physical, emotional, and social effects [25,26]. The prevalence of urinary incontinence after a pelvic surgery for cancer, ranges between 24–52.4% [27,28], with no reported differences regarding the primary cancer site. 

There are two categories of patients with urethral meshes: (1) patients with mesh who may develop a pelvic cancer, and (2) patients with a pelvic cancer and post-surgical urinary incontinence treated with urethral meshes.

For the first category of the patients, in addition to highlighting tumour invasion into adjacent structures, MRI is able to highlight how the cancer affects the urethral mesh, its location, structure and possible complications (Figure 6, Figure 7 and Figure 8).

For the second category of patients treated for a pelvic cancer, MRI is usually used to detect tumour recurrence and short- and long-term mesh complications.

## 5. Urethral Mesh Complications

Complications associated with urethral mesh procedures can occur, but their frequency can vary depending on several factors, including the specific procedure, patient characteristics (including the onset of a pelvic cancer), and the experience of the surgeon. It is important to note that the complication rates can vary depending on the length of follow-up and the definition and reporting of complications across different studies [29]. However, it is generally accepted that two main types of complications can occur: short-term and long-term, both of which also affect cancer patients.

### 5.1. Short-Term Complications

Short-term complications include intraoperative and immediate post-operative complications related to injuries during surgery, acute urinary retention and infections [30,31]. Depending on the mesh pathway we may encounter bladder, urethral or bowel injuries in up to 24% of the TVT, and obturator nerve and artery lesions in up to 2% of the TOT patients [32] (Figure 9, Figure 10 and Figure 11).

The reported rates of acute urinary retention (AUR) following urethral mesh placement have ranged from between 2% to 12% [33,34,35]. However, it’s important to remember that these rates can vary depending on the specific procedure, patient population, the follow-up period of the study and different causes. Several factors can contribute to the development of AUR, such as post-operative swelling, overactive bladder, temporary dysfunction of the bladder muscle or nerves, excessive mesh tension or malpositioned tape sitedtoward the urethra-vesical junction. It is worth mentioning that AUR is often a transient complication and can be managed with conservative measures, the temporary use of a catheter and only in a minority of cases is surgery required.

It is important to note that the occurrence of infection can be influenced by factors such as the patient’s overall health, pre-existing conditions, adherence to sterile techniques during surgery, and post-operative care.

The reported rates of infections in the early post-operative period following urethral mesh procedures have ranged from less than 1% to around 10% [17,36]. Infections can manifest as urinary tract infections (UTIs) or surgical site infections. UTIs are more common, present with symptoms such as increased urinary frequency, urgency, burning sensation during urination, lower abdominal/ pelvic discomfort and have a conservative treatment. Surgical site infections can cause redness, swelling, warmth, pain, or discharge at the incision site and usually require imaging to assess the depth and the local extent of the disease (Figure 12).

Inflammation and infections of urethral meshes are easily depicted on thin T2WI slices, as having thickened uni- or bilateral limbs with high signal intensity. Abscesses may affect any portion of the mesh and therefore requires evaluation of the trajectory of the mesh in all planes. Infected or inflammed mesh appears with avid enhancement on post-contrast sequences, helping identifing the sinus trast and abscesses (Figure 13 and Figure 14).

### 5.2. Long-Term Complications

Delayed-postoperative complications range from 3–25% of cases and include bladder outlet obstruction, urinaryx incontinence and recurrent urinary tract infections in the majority of the cases; while only 2–3% of these are related to mesh integrity and/or abnormal localization [17,36,37]. Of the latter, the most common complications are mesh exposure/erosion and extrusion, and may lead to reccurent urinary infections or irritative pelvic simptoms. Whenever a mesh lesion is clinically suspected, we suggest performing a dedicated MRI for assessing mesh integrity, while also acknowledging the potential utility of ultrasound.

Erosion or exposure of the mesh usually involve vagina, urethra or bladder, and are recognised when the mesh limbs are seen within the lumen of these sturctures [16]. The involved mesh limb will be thicknened with high T2WI signal intensity and mild- to avid- enhancement on post contrast sequences. 

In many cases, inflammation and infection occurring late after surgery secondary to mesh erosion, particularly involving the vagina. Surgery may be needed to remove the affected limb or entire mesh or correct other complications, although the overall incidence is rare. Over 50% of women who experienced erosion with non-absorbable synthetic mesh needed to have the mesh surgically removed. Some patients required two or more operations after the mesh was removed (Figure 15, Figure 16, Figure 17, Figure 18 and Table 1).

## 6. Conclusions

In this review, we have discussed the various complications associated with urethral mesh placement in oncologic patients. The potential advantage of urethral mesh lies in its ability to provide effective support for weakened tissues, addressing conditions like stress urinary incontinence, which may affect oncological patients, although complications such as inflammation, mesh exposure, or bladder erosion pose significant disadvantages, emphasizing the importance of careful patient selection and thorough evaluation. From normal appearance to intraoperative and immediate post-operative complications and delayed complications, it is evident that a thorough understanding of these complications is essential for clinicians managing such patients. By recognizing the pathological aspects and employing appropriate preventive strategies, clinicians can optimize patient outcomes and minimize the incidence of complications related to urethral mesh placement. MRI remains the imaging modality of choice in assessing the integrity and location of urethral meshes.

## Figures and Tables

**Figure 1 cancers-15-05599-f001:**
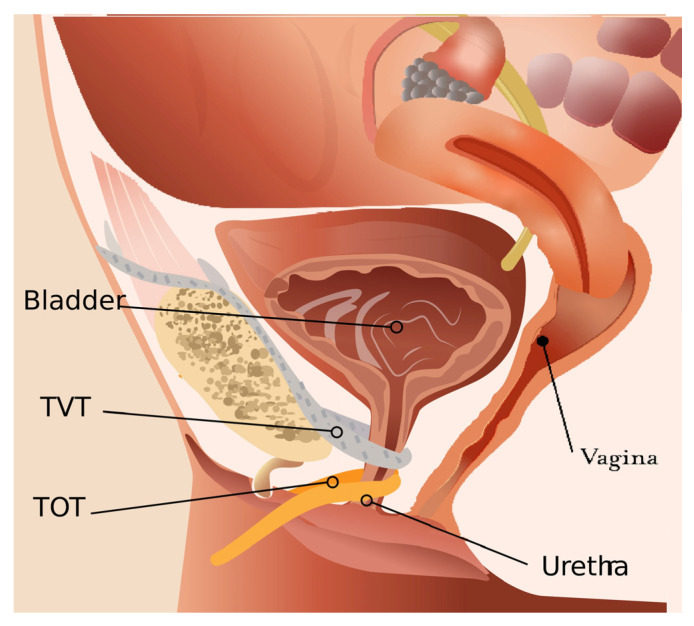
TVT and TOT urethral meshes on sagittal plane.

**Figure 2 cancers-15-05599-f002:**
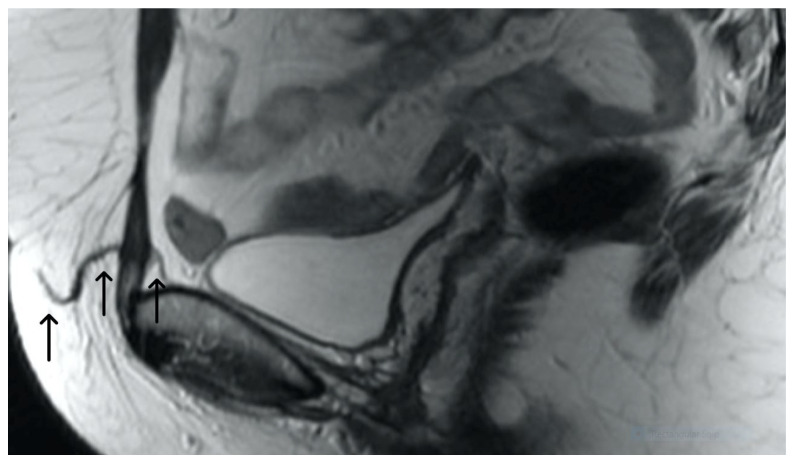
Normal appearance of TVT mesh. Sagittal T2WI showing a thin, linear low signal intensity mesh (arrows), anchor within the subcutaneous fat, extending through the rectus sheath and inferior to the bladder within the retropubic space.

**Figure 3 cancers-15-05599-f003:**
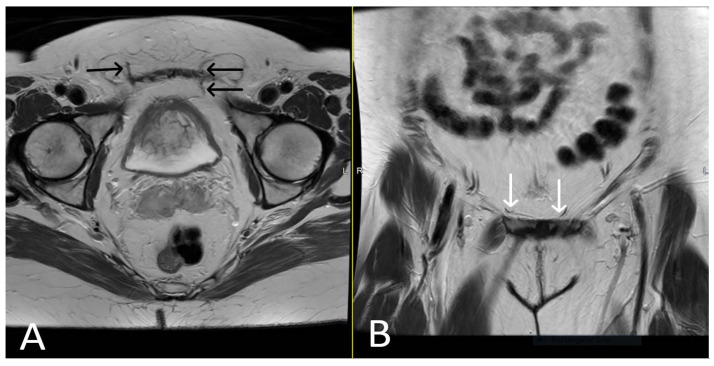
Normal appearance of TVT mesh. Axial (**A**) and coronal (**B**) T2WI demonstrating linear low signal intensity mesh anchor within the subcutaneous fat, extending through the rectus sheath (black arrows) and anteroinferior to the bladder within the retropubic space (white arrows).

**Figure 4 cancers-15-05599-f004:**
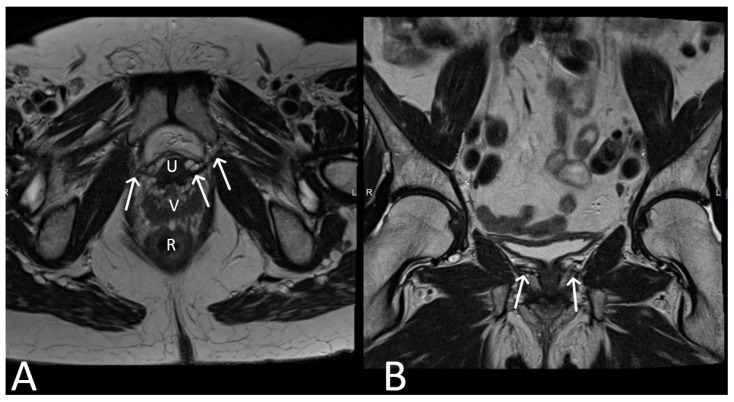
Normal appearance of TOT mesh. Axial (**A**) and coronal (**B**) T2WI display a linear low signal intensity of a normal TOT mesh within the paraurethral space, posterior to the mid-urethra (U), anterior to vagina (V) and rectum (R), extending beyond the obturator foramen, towards the adductor muscles (white arrows).

**Figure 5 cancers-15-05599-f005:**
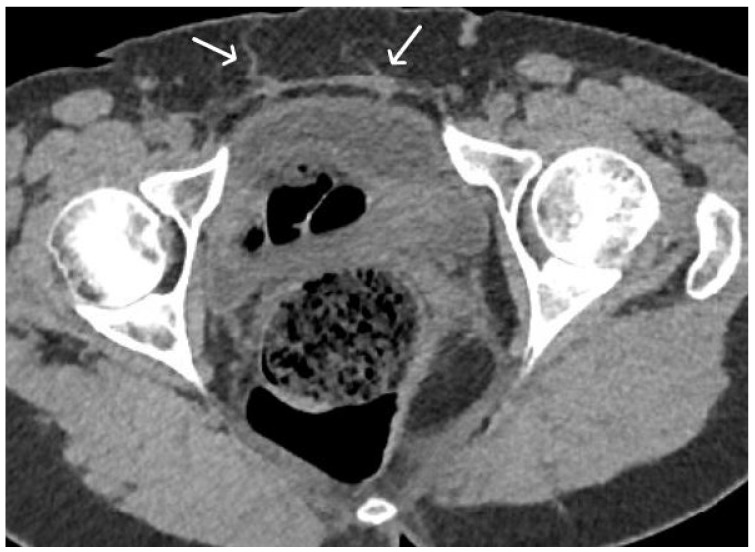
Normal TVT mesh appearance on CT imaging. The mesh is seen as a thin and linear structure passing through the anterior abdominal wall (arrows).

**Figure 6 cancers-15-05599-f006:**
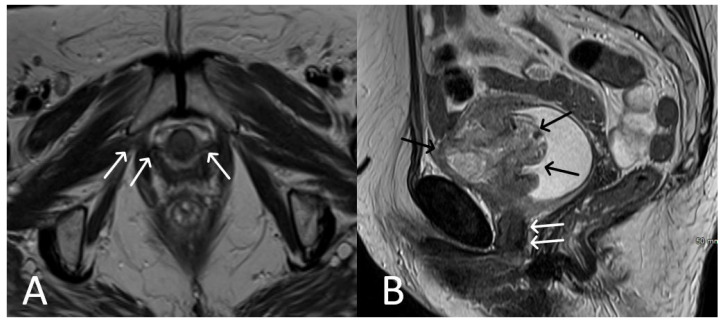
Patient with bladder cancer and TOT mesh. Axial (**A**) and sagittal (**B**) T2WI images demonstrate extensive abnormal soft tissue of the anterior aspect of bladder (black arrows) in the setting of a normal TOT (white arrows).

**Figure 7 cancers-15-05599-f007:**
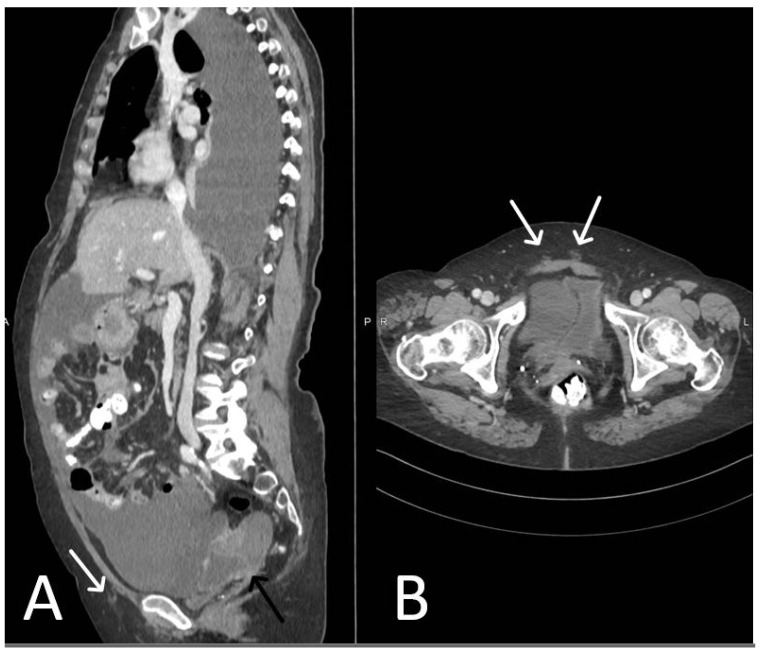
Patient with ovarian cancer and TVT mesh. Sagittal (**A**) and axial (**B**) contrast enhanced CT showing a mixed cystic-solid ovarian mass (black arrow), ascites and pleural effusion. Normal TVT is visible in both planes (white arrows).

**Figure 8 cancers-15-05599-f008:**
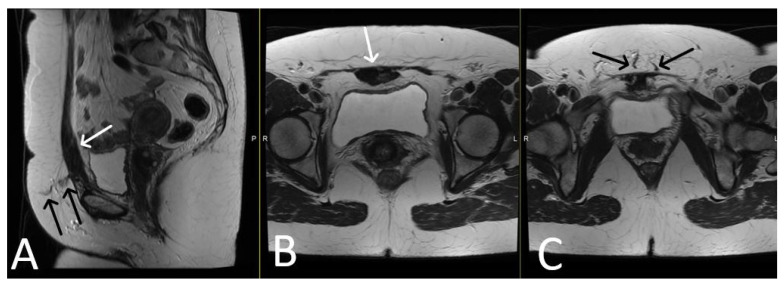
Desmoid tumour of the anterior abdominal wall, with TVT mesh. Sagittal (**A**) and axial (**B**,**C**) T2WI demonstrate low T2 signal desmoid tumour of the anterior abdominal wall (white arrows). Normal appearance of the TVT seen passing through the inferior aspect of the desmoid tumour (black arrows).

**Figure 9 cancers-15-05599-f009:**
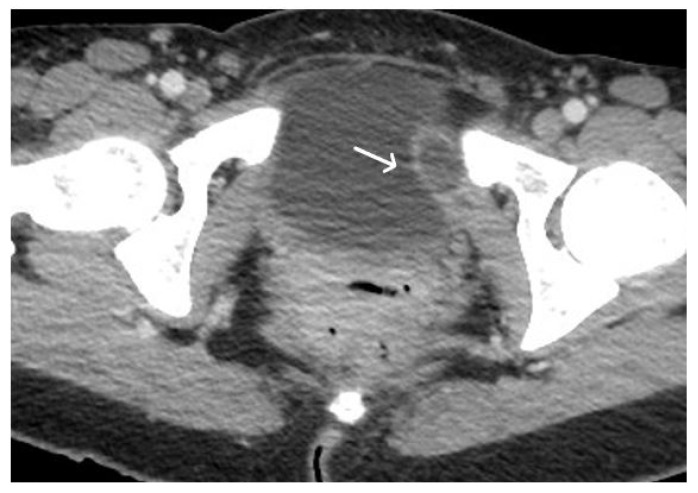
Acute haematoma post TVT procedure. Post-contrast CT imaging is highlighting a well-defined collection (white arrow) in left obturator internus muscle; no signs of active bleeding are noted.

**Figure 10 cancers-15-05599-f010:**
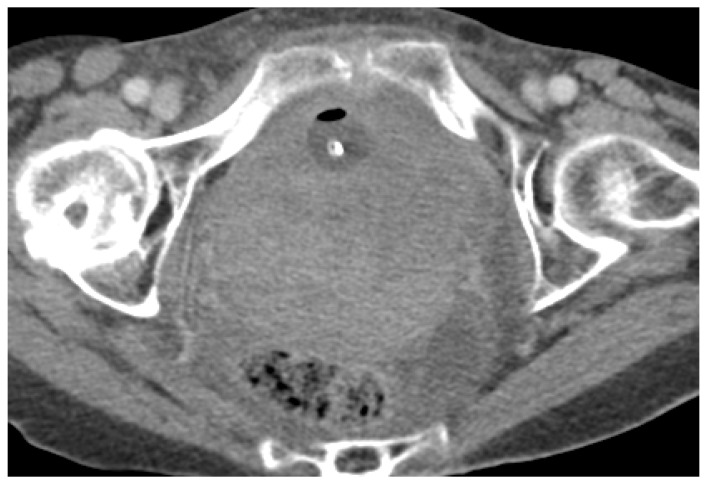
Acute pelvic haematoma post sacrohysteropexy. Post-contrast CT imaging demonstrates a large, well-defined and hyperdense collection (haematoma) located within the pelvis, with important mass effect on the bladder and rectum; no signs of active bleeding are noted.

**Figure 11 cancers-15-05599-f011:**
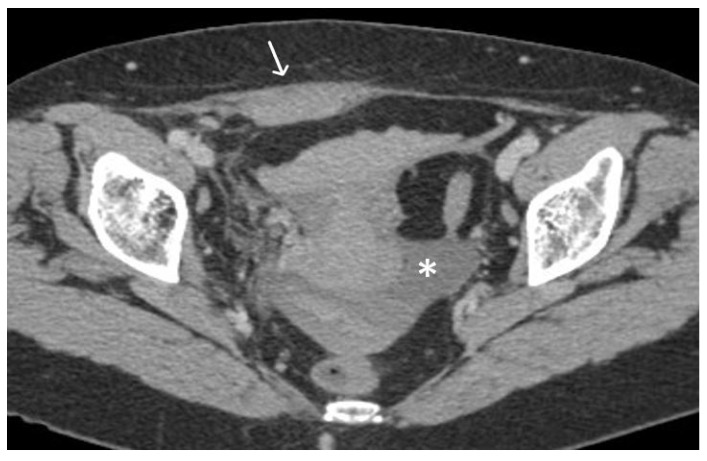
Haemoperitoneum and right rectus sheath haematoma post sacro-hysteropexy. Post-contrast CT imaging demonstrates hyperdense free fluid (*) located in the pelvic peritoneal reflection suggestive of haemoperitoneum and a well-defined hyperdense collection (white arrow) located within the right rectus abdominal muscle consistent with a haematoma; no signs of active bleeding are noted.

**Figure 12 cancers-15-05599-f012:**
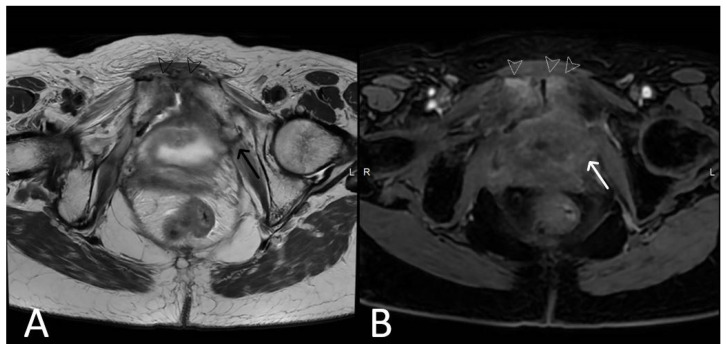
History of cervical cancer, TOT in situ and pubic symphysis osteomyelitis. Axial T2WI (**A**) and T1 fat saturated, post contrast (**B**) sequences demonstrate thickened left limb of TOT traversing obturator foramen to adductor muscles (black arrow), with post-contrast enhancement (white arrow) sugestive of infection. Enhancement at the pubic symphysis (white arrowheads) with loss of normal low T1 and T2 signal of the cortex (black arrowheads) consistent with osteomyelitis.

**Figure 13 cancers-15-05599-f013:**
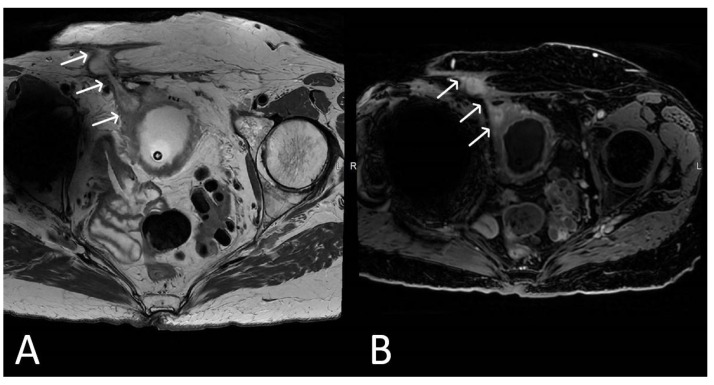
*Upper images*: Complicated, infected right limb of TVT and bladder erosion. Axial T2WI (**A**) and T1 fat saturated, post contrast (**B**) sequences demonstrate TVT with abscess of right limb, extending from the bladder wall to the skin surface in keeping with a sinus tract (white arrows) due to a bladder erosion. *Lower images*: Complicated, infected right limb of TVT. Coronal (**C**) and axial (**D**) T2WI sequences demonstrate thickened right limb of TVT within bladder wall (white arrows). Extension to skin partially visualised on axial sequence (black arrow). Artefact from right hip prosthesis.

**Figure 14 cancers-15-05599-f014:**
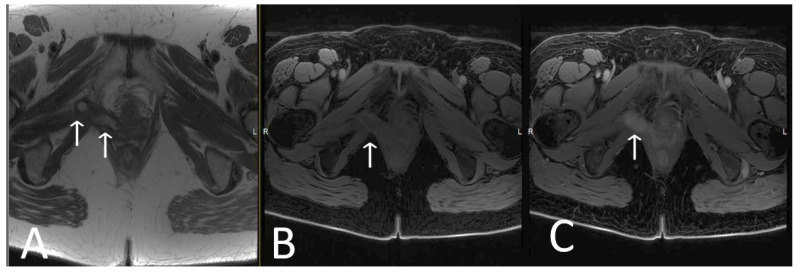
Infected limb of TOT and vaginal exposure. Axial T2WI (**A**), T1 fat saturated (**B**), and T2 fat saturated, post contrast (**C**) sequences of TOT demonstrate marked thickening and avid post contrast enhancement of infected right limb of mesh (white arrows), most likely due to a vaginal exposure.

**Figure 15 cancers-15-05599-f015:**
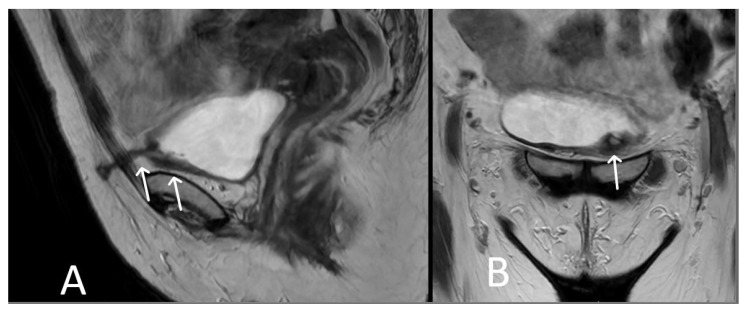
Bladder erosion. Sagittal (**A**) and coronal (**B**) T2WI showing left arm of TVT passing through the left anterolateral bladder wall (white arrows). The bladder wall is markedly thickened at this site with irregularity and oedema of the adjacent urothelium.

**Figure 16 cancers-15-05599-f016:**
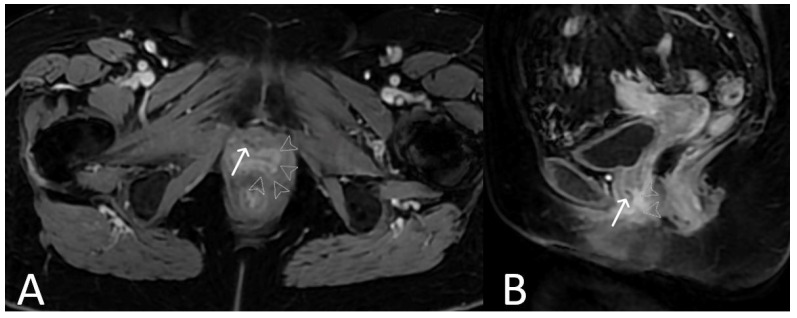
Urethral erosion. Axial (**A**) and sagittal (**B**) T1 fat saturated post-contrast sequences, demonstrating linear low signal within urethra (white arrows), reflecting erosion of mesh into the urethra. There is enhancement of the lower urethra and vagina (arrowheads). Cystoscopy confirmed synthetic fibres within urethral lumen, which were resected.

**Figure 17 cancers-15-05599-f017:**
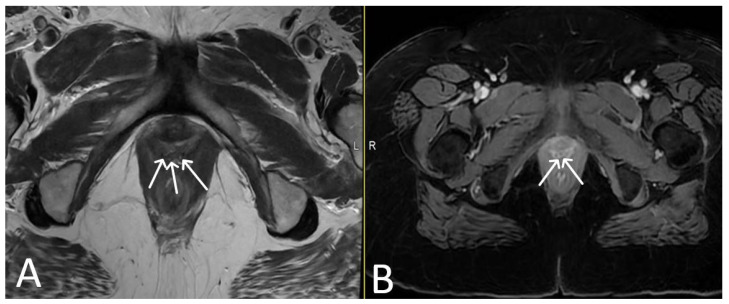
Vaginal exposure. (Same patient). Axial T2WI (**A**) and axial fat saturated post-contrast T1 (**B**) images. Linear low signal intensity mesh within the oedematous vagina, compatible with mesh exposure (white arrows).

**Figure 18 cancers-15-05599-f018:**
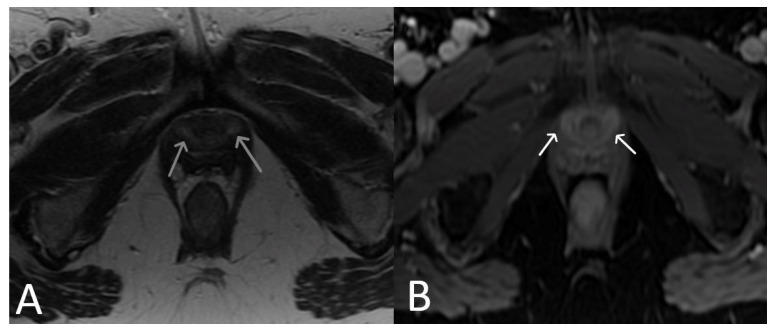
Normal clinical examination. Inflammation without mesh erosion. Axial T2WI (**A**) and axial fat saturated post-contrast T1WI (**B**) demonstrating abnormal high/intermediate T2 signal intensity of mesh as it passes between the vagina and urethra (grey arrows). On post contrast imaging, this portion of the mesh demonstrates avid enhancement (white arrows).

**Table 1 cancers-15-05599-t001:** Short and long-term complications of urethral meshes.

Short-Term			Long-Term
	TVT	TOT	TVT and TOT
Injuries	Bladder	Obturator nerve	Bladder outlet syndrome
	Urethra	Obturator vessels (artery and vein with medial and lateral branches)	Urge-incontinence Recurrent urinary infections Acute urinary retention Mesh exposure Mesh extrusion Mesh infection
	Bowel		
	Vessels		
Acute urinary retention			
Mesh Infections			

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
