# Peer review of "Urethral Mesh Assessment in Cancer Patients"

_cancers, 2023, doi:10.3390/cancers15235599_

Round 1
Reviewer 1 Report
Comments and Suggestions for Authors
Urethral mesh assessment in cancer patients
It as a very interesting and original paper however, it has some major flaws.
Key words
Please change the MeSH term – there is no such thing as mesh cancer.
“ Although mesh surgery does not increase the risk of cancer [4], women may still 35 develop pelvic cancer, or require urethral mesh secondary to cancer surgery.” – instead of using the cord cancer it is better to use the word malignancy. Not all malignancies are cancers.
Figure 10 and 11– irrelevant. I do not see any point in presenting the figure of the complications after the POP procedure. I guess the authors want to present all they have instead of keeping to the topic.
Is there a “tape migration to-189 ward the urethra-vesical junction.” – it is just bad placement of the tape. The tape does not migrate.
What do you mean by bladder irritability – there is no such term in the terminology.
“The reported rates of infections in the early post-operative period following urethral 198 mesh procedures have ranged from less than 1% to around 10%” – the sentence is repeated
Figure 14 – there is no such term as vaginal erosion – it is highly wrong and a mistake. Exposition of the tape to the vagina.
Urge incontinence – old term
Do not use “mesh erosion”.
Whenever a mesh lesion is clinically suspected, a dedicated 240 MR should be performed to assess the mesh integrity. – it is too strongly recommended – on what recommendation do the authors base on?. An ultrasound is also a good examination.
Presenting the Figure just after you said about exposition of the mesh is misleading.
Table 1
Why only obturator artery?
MRI remains the imaging modality of 281 choice in assessing the integrity and location of urethral meshes. – it is not true. An ultrasound imagining is the first choice – based on the costs and availability.
Author Response
R1
It as a very interesting and original paper however, it has some major flaws. à We appreciate your time and efforts.
- Key words
Please change the MeSH term – there is no such thing as mesh cancer à We replaced the term with “urethral neoplasm”
- “Although mesh surgery does not increase the risk of cancer [4], women may still 35 develop pelvic cancer, or require urethral mesh secondary to cancer surgery.” – instead of using the cord cancer it is better to use the word malignancy. Not all malignancies are cancers à We replaced the word with malignancies. Thank you
- Figure 10 and 11– irrelevant. I do not see any point in presenting the figure of the complications after the POP procedure. I guess the authors want to present all they have instead of keeping to the topic. à We aimed to present the imaging appearances of acute bleeding, hematoma and hemoperitoneum that looks similar in both types of patients. Of course, if the reviewers believe the images are not useful, we are happy to cut them out.
- Is there a “tape migration toward the urethra-vesical junction.” – it is just bad placement of the tape. The tape does not migrate. à We rephrased it as: malpositioned tape sited towards the …”
We are sorry for using the “migration” word, we believed it is acceptable, as it was mentioned in previous papers; please find below the PubMed papers that refers to the tape migration in both males and females: https://pubmed.ncbi.nlm.nih.gov/?term=urethral+tape+migration
- What do you mean by bladder irritability – there is no such term in the terminology à Thank you and sorry for that. We replaced the term with “overactive bladder”.
- “The reported rates of infections in the early post-operative period following urethral mesh procedures have ranged from less than 1% to around 10%” – the sentence is repeated à
- Figure 14 – there is no such term as vaginal erosion – it is highly wrong and a mistake. Exposition of the tape to the vagina. Do not use “mesh erosion”. à We replace the word with “exposure”, and added the reference by Haylen BT et al related to the terminology.
Ref: Haylen BT, Freeman RM, Swift SE, Cosson M, Davila GW, Deprest J, Dwyer PL, Fatton B, Kocjancic E, Lee J, Maher C, Petri E, Rizk DE, Sand PK, Schaer GN, Webb R (2011) An International Urogynecological Association (IUGA)/International Continence Society (ICS) joint terminology and classifcation of the complications related directly to the insertion of prostheses (meshes, implants, tapes) and grafts in female pelvic foor surgery. Neurourol Urodyn 30 (1):2-12. doi:10.1002/nau.21036
- Urge incontinence – old term à Replaced with “urinary incontinence”
- Whenever a mesh lesion is clinically suspected, a dedicated MR should be performed to assess the mesh integrity. – it is too strongly recommended – on what recommendation do the authors base on? An ultrasound is also a good examination. à Thank you for the suggestion. Indeed, there are no clear recommendations about imaging, and the guidelines state that “Diagnostic testing for a suspected mesh complication can include cystoscopy, proctoscopy, colonoscopy, or radiologic imaging”. It is our institutional protocol that favors MRI; we rephrased and toned down the sentence as follows: Whenever a mesh lesion is clinically suspected, we suggest performing a dedicated MRI for assessing mesh integrity, while also acknowledging the potential utility of ultrasound.
- Presenting the Figure just after you said about exposition of the mesh is misleading à We moved the paragraph before the images. Please advise us on this matter.
- Table 1 Why only obturator artery? à We added obturator vessels (artery and vein with medial and lateral branches)
- MRI remains the imaging modality of choice in assessing the integrity and location of urethral meshes. – it is not true. An ultrasound imagining is the first choice – based on the costs and availability à We completely agree with the reviewer in non-oncological patients; however, in our institution MRI is the preferred modality, especially in oncological patients. Furthermore, based on our experience, we consider MR is more suitable in oncological patients, and we will be carefully to state it accordingly within the manuscript. We added a new paragraph: “Existing guidelines recognize the usefulness of imaging methods in identifying and characterizing urethral mesh complications, although they do not specify which imaging method should be used first. Ultrasound and dedicated MRI of the pelvis should be considered. In our institution, non-oncology patients with symptoms suggestive of urethral mesh complications will be referred for ultrasound. If this is not able to fully characterize the suspected lesion, patients are referred for MRI.
In the case of oncological, symptomatic patients, MRI is used of first intention in our institution. A dedicated MR is crucial in assessing the exact site and extent of mesh related complications. Particularly in patients with known exposed or extruded mesh, MRI provides information regarding the remainder of the mesh that is inaccessible to clinical examination or ultrasound.”
Reviewer 2 Report
Comments and Suggestions for Authors
The review article investigated normal magnetic resonance imaging (MRI) and computed tomography (CT) mesh appearances and highlighted the pathological aspects associated with urethral mesh complications including both short-term and long-term post-operative complications. Following are some concerns that the authors need to revise and address before considering publication.
1. The authors should add the research gap in the introduction.
2. Please add references to support your evidence in lines 55-57, 75-118, 135-137, 187-197, 199-217, 238-265.
3. The conclusion should highlight the potential advantages or disadvantages of urethral mesh.
Author Response
The review article investigated normal magnetic resonance imaging (MRI) and computed tomography (CT) mesh appearances and highlighted the pathological aspects associated with urethral mesh complications including both short-term and long-term post-operative complications. Following are some concerns that the authors need to revise and address before considering publication.
- The authors should add the research gap in the introduction à We added the following:
The history of urethral meshes traces a journey from ancient civilizations' use of natural materials like animal ligaments to reinforce weakened tissues, to the mid-20th-century breakthrough of synthetic meshes, notably polypropylene, revolutionizing urological surgery, especially in treating stress urinary incontinence (SUI) [B]. These synthetic mesh tapes, while widely used, posed challenges due to proinflammatory responses after implantation [C]. The advent of the tension-free vaginal tape (TVT) marked a significant milestone, yet blind access during TVT procedures led to complications, necessitating routine cystoscopy [10]. In 2001, second-generation slings emerged, like transobturator meshes (TOT), offering safer alternatives [11]. The latest innovation, the mini sling or single-incision sling (SIS), introduced a minimally invasive approach, involving a single vaginal incision, reducing surgical invasiveness, potentially improving recovery, and minimizing post-operative discomfort [D,E].
B- Afonso JS, Jorge RM, Martins PS, Soldi Mda S, Alves OL, Patricio B, Mascarenhas T, Sartori MG, Girao MJ. Structural and thermal properties of polypropylene mesh used in treatment of stress urinary incontinence. Acta Bioeng Biomech. 2009;11(3):27-33
C - Nolfi AL, Brown BN, Liang R, Palcsey SL, Bonidie MJ, Abramowitch SD, Moalli PA. Host response to synthetic mesh in women with mesh complications. Am J Obstet Gynecol. 2016 Aug;215(2):206.e1-8. doi: 10.1016/j.ajog.2016.04.008.
D - Abdel-Fattah M, Cooper D, Davidson T, Kilonzo M, Hossain M, Boyers D, Bhal K, Wardle J, N'Dow J, MacLennan G, Norrie J. Single-Incision Mini-Slings for Stress Urinary Incontinence in Women. N Engl J Med. 2022 Mar 31;386(13):1230-1243. doi: 10.1056/NEJMoa2111815. = 5
E - Moore RD, Serels SR, Davila GW, Settle P. Minimally invasive treatment for female stress urinary incontinence (SUI): a review including TVT, TOT, and mini-sling. Surg Technol Int. 2009 Apr;18:157-73 Top of Form
- Bottom of Form
- Please add references to support your evidence in lines 55-57, 75-118, 135-137, 187-197, 199-217, 238-265. à Thank you for this suggestion. We mentioned already used references and added some new ones as well:
- lines 55-57 – Ref. F, G, H:
Manganaro L, Lakhman Y, Bharwani N, Gui B, Gigli S, Vinci V, Rizzo S, Kido A, Cunha TM, Sala E, Rockall A, Forstner R, Nougaret S. Staging, recurrence and follow-up of uterine cervical cancer using MRI: Updated Guidelines of the European Society of Urogenital Radiology after revised FIGO staging 2018. Eur Radiol. 2021 Oct;31(10):7802-7816. doi: 10.1007/s00330-020-07632-9
Colombo N, Preti E, Landoni F, Carinelli S, Colombo A, Marini C, Sessa C; ESMO Guidelines Working Group. Endometrial cancer: ESMO Clinical Practice Guidelines for diagnosis, treatment and follow-up. Ann Oncol. 2013 Oct;24 Suppl 6:vi33-8. doi: 10.1093/annonc/mdt353
Glynne-Jones R, Wyrwicz L, Tiret E, Brown G, Rödel C, Cervantes A, Arnold D; ESMO Guidelines Committee. Rectal cancer: ESMO Clinical Practice Guidelines for diagnosis, treatment and follow-up. Ann Oncol. 2017 Jul 1;28(suppl_4):iv22-iv40. doi: 10.1093/annonc/mdx224
-75-118 and 135-137 – Ref I:
Jung BC, Tran NA, Verma S, Dutta R, Tung P, Mousa M, Hernandez-Rangel E, Nayyar M, Lall C. Cross-sectional imaging following surgical interventions for stress urinary incontinence in females. Abdom Radiol (NY). 2016 Jun;41(6):1178-86. doi: 10.1007/s00261-016-0684-0
- 187-197 - Ref J
Hammett J, Peters A, Trowbridge E, Hullfish K. Short-term surgical outcomes and characteristics of patients with mesh complications from pelvic organ prolapse and stress urinary incontinence surgery. Int Urogynecol J. 2014 Apr;25(4):465-70. doi: 10.1007/s00192-013-2227-3.
- 199-217 – Ref 24-26
- 238-265 – Ref 29,30 and we added Ref A:
- The conclusion should highlight the potential advantages or disadvantages of urethral mesh à We added in conclusion: The potential advantage of urethral mesh lies in its ability to provide effective support for weakened tissues, addressing conditions like stress urinary incontinence, which may affect oncological patients, although complications such as inflammation, or mesh erosion pose significant disadvantages, emphasizing the importance of careful patient selection and thorough evaluation.
Reviewer 3 Report
Comments and Suggestions for Authors
This articles entitled „Urethral mesh assessment in cancer patients“ presented actually review of urethral mesch in cancer patients, with short introduction and very short dsicussion and conclusion. The presenters presented the given objectives of the work with numerous pictorial presentations and comments in the components of the article, along with a very short discussion of similar research and citations of adequate fresh literature.
However, structurally, the articles are not adequate in this form:
The introduction is too short, and the comments on 17 images do not fully support the review as a global presentation of meta-analyses on the mentioned problem, which is clinically interesting and deserves the attention of the publication.
Therefore, I recommend the authors to deepen the discussion and discuss the problem by citing the literature as it is done in review articles. Otherwise, this is a very brief pictorial representation of the problem, but not for this magazine, but Images in medicine...
1. What is the main question addressed by the research?
Described in Title and Methodology
2. Do you consider the topic original or relevant in the field? Does it address a specific gap in the field?
Yes
3. What does it add to the subject area compared with other published material?
Clinical problem
4. What specific improvements should the authors consider regarding the methodology? What further controls should be considered?
No need
5. Are the conclusions consistent with the evidence and arguments presented and do they address the main question posed?
Yes
6. Are the references appropriate?
Yes
7. Please include any additional comments on the tables and figures.
In my comments
Author Response
This articles entitled „Urethral mesh assessment in cancer patients“ presented actually review of urethral mesch in cancer patients, with short introduction and very short dsicussion and conclusion. The presenters presented the given objectives of the work with numerous pictorial presentations and comments in the components of the article, along with a very short discussion of similar research and citations of adequate fresh literature.
However, structurally, the articles are not adequate in this form:
The introduction is too short, and the comments on 17 images do not fully support the review as a global presentation of meta-analyses on the mentioned problem, which is clinically interesting and deserves the attention of the publication. à We appreciate your comment. We added more information in the introduction and within the manuscript; and we changed the order of some pictures. We hope this present form of interest for you. We would like to mention that this invited paper was intended to be a pictorial and not a systematic review, due to the lack of publications regarding mesh complications, especially in oncological patients.
Therefore, I recommend the authors to deepen the discussion and discuss the problem by citing the literature as it is done in review articles. Otherwise, this is a very brief pictorial representation of the problem, but not for this magazine, but Images in medicine. à Thank you for your comment. As suggested by the other reviewer, we added more data supported by references within the manuscript (Ref A à K).
Reviewer 4 Report
Comments and Suggestions for Authors
This paper has no real purpose and it is not a systematic review or meta-analysis.
It does not present a flowchart PRISMA and does not address existing published literature, but reports images whose origins are unknown.
This article adds nothing new to the literature.
Comments on the Quality of English LanguageMinor editing of English language required.
Author Response
This paper has no real purpose and it is not a systematic review or meta-analysis.
It does not present a flowchart PRISMA and does not address existing published literature, but reports images whose origins are unknown.
This article adds nothing new to the literature.
- Thank you for your time; first, this invited paper is a pictorial review and not a systematic/meta-analysis, and of course we are aware of the differences (perhaps it was stated in your invitational letter) but it was never intended to be a systematic review, due to the lack of publications regarding this matter;
- Second, the reviewer may not have been able to thoroughly grasp the nuances of the content. The scarcity of published articles focusing on female urethral mesh imaging appearances indicates a significant gap in the existing literature. Additionally, the limited available research makes it challenging to assess these appearances accurately, emphasizing the complexity of the subject matter. Furthermore, there is a notable absence of published papers addressing mesh complications specifically in cancer patients. Given these gaps, the study’s novelty becomes apparent (to the best of our knowledge, this stands as a first paper).
Round 2
Reviewer 1 Report
Comments and Suggestions for Authors
Introduction
Instead of urethral mesh – urethral tapes or proper term would be mid-urethral sling (MUS) instead of urethral mesh through the manuscript.
Please change the sentence” blind access during TVT procedures led to complications, necessitating routine cystoscopy [9].”
In the cookbook, in the procedure was a cystoscopy – not because of many complications that led to this, but because of the anatomic situation in the pelvis.
TOT is not a second-generation sling – it is just another route of positioning the tape that was developed later.
“The latest innovation, the mini sling or single-incision sling (SIS), introduced a minimally invasive approach, involving a single vaginal incision, reducing surgical invasiveness, potentially improving recovery, and minimizing post-operative discomfort [D,E].” – that is not what is true according to the literature.
The authors should have corrected the references; introducing alphabetical order as additional is improper.
In the paragraph about the tapes, the information about TOT and TVT is repeated from the introduction.
“However, even if mini-slings aim to achieve similar success rates, their short- and long-term efficacy and complications have yet to be established [13,14].” – it is not true. It is already in the literature.
Is it inappropriate to say “cancer patients”.
“The prevalence of urinary incontinence after a pelvic surgery for cancer ranges between 24 - 52.4% [20,21], with no reported differences regarding the primary cancer site.” - it is not true – previously, the authors said that pelvic cancer would be all cancers in the pelvis. After that you refer only to gynecological and cervical cancer.
The paragraph – “urethral mesh complications” – it does not present any real value, repeated general sentences.
Again – the authors did not remove the image of the patient after sacropexy (Figure 10 and 11) as I suggested – it is not the article's topic.
Overactive bladder is not the reason for acute urinary retention.
Reviewer 3 Report
Comments and Suggestions for Authors
Without comments
Comments on the Quality of English LanguageMinor editing of English language required
Reviewer 4 Report
Comments and Suggestions for Authors
Good
Comments on the Quality of English LanguageGood